# READING TEA LEAVES FOR DE NOVO PROTEIN DESIGN

**Lorenzo Pantolini**
Biozentrum, University of Basel
SIB Swiss Institute of Bioinformatics
lorenzo.pantolini@unibas.ch

**Janani Durairaj**
Biozentrum, University of Basel
SIB Swiss Institute of Bioinformatics
janani.durairaj@unibas.ch

## ABSTRACT

*De novo* protein design expands the functional protein universe beyond natural evolution, offering vast therapeutic and industrial potential. Monte Carlo sampling in protein design is under-explored due to the typically long simulation times required or prohibitive time requirements of current structure prediction oracles. Here we make use of a 20-letter structure-inspired alphabet derived from protein language model embeddings to score random mutagenesis-based Metropolis sampling of amino acid sequences. This facilitates fast template-guided and unconditional design, generating sequences that satisfy *in silico* designability criteria without known homologues. Ultimately, this unlocks a new path to fast and *de novo* protein design.

## 1 INTRODUCTION

Computational protein design has advanced rapidly in recent years, largely tracking successes in deep learning-based structure prediction (Khakzad et al., 2023). The design challenge is frequently framed as a discrete optimisation problem: the minimisation of a loss function, $\mathcal{L}(x)$, over a sequence $x \in A^N$, where $A$ is the set of amino acids and $N$ the length. Here, $\mathcal{L}$ proxies protein "quality" (e.g., stability, solubility, folding propensity etc.). However, the design space $A^N$ is vast, with typical quality estimation functions such as those derived from AlphaFold2 (Jumper et al., 2021) structure prediction remaining too slow for effective exploration. Consequently, generative approaches like RFDiffusion (Watson et al., 2023) and gradient-based methods like BindCraft (Pacesa et al., 2025) have emerged to circumvent the lack of a fast, informative loss function with good optimisation properties.

The Embedded Alphabet (TEA) (Pantolini et al., 2025) was recently introduced, providing a novel means of representing proteins by leveraging embeddings from protein language models (pLMs) to convert amino acid sequences into sequences of a new 20-letter alphabet, effectively extracting latent structural information. In essence, TEA conversion has minimal computational cost but is still able to tell if two proteins share a similar fold, irrespective of their amino acid sequence similarity. In addition, the Shannon entropy of its probability vectors correlates with structural confidence.

Markov chain Monte Carlo (MCMC) is attractive for its simplicity but struggles when energy oracles provide poor signal or are computationally expensive. Previous pLM-based MCMC attempts faced significant speed-accuracy trade-offs: contact map oracles required hundreds of thousands of steps (Verkuil et al., 2022), while explicit structure prediction like ESMFold (Lin et al., 2022) were too slow per step (Hie et al., 2022; Lála et al., 2025). In this work we propose TEA-LEAVES, a framework where we exploit the synergy between TEA-encoded and amino acid probabilities. By jointly optimising $p(\text{structure}|\text{sequence})$ and $p(\text{sequence})$, TEA-LEAVES converges to diverse minima orders of magnitude faster than prior attempts.

Overall, we obtain high quality *de novo* solutions that pass established *in silico* designability criteria across various artificial and natural scaffolds. We simulate a diverse landscape of novel folds topologies and secondary structures. Thus, extracting discrete structural information from pLM embeddings may provide a novel, rapid approach to protein design.

Figure 1: **Overview A)** Monte Carlo sampling procedure, where at each step a random mutation is accepted with probability $P_a(\Delta E)$. **B)** Depiction of the energy function inputs as described in Section 2.

## 2 TEA ENERGY FUNCTIONS ENABLE FAST MCMC CONVERGENCE

To generate sequences, we employ an energy-based Markov Chain Monte Carlo (MCMC) simulation, utilising the Metropolis algorithm. The simulation begins with an initial randomly generated sequence of length $N$, denoted as $x$. At each iteration, a random residue position $x_i$ is selected and undergoes a random mutation to a different amino acid chosen with uniform probability, resulting in a candidate sequence $x'$. The transition from the current state to the candidate state is regulated by the energy difference, $\Delta E = E(x') - E(x)$, with the acceptance probability $P_a$ defined as:

$$P_a(\Delta E) = \min\left[1, e^{-\frac{\Delta E}{T}}\right] \tag{1}$$

We define four energy terms for an amino acid sequence $x$ and its corresponding TEA sequence $\tau : x \to x^\tau$, namely $E_{LM}(x)$, $E_T(x^\tau)$, $E_H(x^\tau)$, and $E_D(x^\tau)$, described in detail below. To quantify agreement between a proposed state and a target distribution, we utilise a generalised cross-entropy objective:

$$E(\mathbf{z}, \mathbf{q}) = -\frac{1}{N}\sum_{i=1}^{N}\sum_{j=1}^{M} z_{ij}\log(q_{ij}) \tag{2}$$

From this term we define two distinct energy terms. The first, $E_{LM}(x)$, approximates the sequence prior $p(x)$ using ESM2 Lin et al. (2023). In this case, $\mathbf{q}$ is obtained by applying a *softmax* function to the ESM2 residue logits generated from $x$, while $\mathbf{z}$ is the one-hot encoding of that same sequence. To accelerate sampling, we compute ESM2 logits ($M = 33$) using a single unmasked forward pass, which we found to correlate 99% with the computationally expensive masked marginals formulation (Salazar et al., 2020), which would require $N$ passes each masking $x_i$ (Supplementary Figure S2). The second term derived from Equation 2 is $E_T(x^\tau)$, which approximates the structural likelihood $p(y = Y|x)$, with $y$ being the 3D structure of $x$ and $Y$ the structure of a template. For this term, the arguments are the residue probabilities $\mathbf{q}^\tau$ derived from the *softmax* of TEA logit vectors ($M = 33$) and the target $\mathbf{z}^\tau$ corresponding to the one-hot encoding of the TEA-template sequence. Structural confidence is estimated via the TEA entropy $E_H$ (Pantolini et al., 2025), which proxies $p(y|x)$:

$$E_H(x^\tau) = -\frac{1}{N}\sum_{i=1}^{N}\frac{1}{\log M}\sum_{j=1}^{M} q_{ij}^\tau \log(q_{ij}^\tau) \tag{3}$$

Finally, to encourage topological variety and discourage secondary structure repetition during unconditional generation, we employ a diversity term $E_D(x^\tau)$, defined as 1 minus the ratio of unique $k$-mers ($k = 3$) in the TEA sequence.

For each specific design task, the total energy is defined as a weighted linear combination of these individual terms, parametrized by the vector of weights $\boldsymbol{\lambda}$. All simulations were run for 30,000 steps with a simulated annealing scheduler starting at a temperature of 0.005 and decaying the temperature by half every 3,000 steps. Both starting temperature and energy weights were empirically determined to balance exploration and convergence and the same values were used for all simulations. We use the 4-bit quantized ESM2 model, as done for training TEA, enabling memory-efficient

sampling of even longer proteins. Batching has sub-linear scaling enabling multiple parallel Markov chains on the same GPU. **Thus, for a protein of length 100, one design takes 24 minutes on an A100-40G GPU node**, while 64 designs batched takes 7 hours (instead of 24 minutes $\times$ 64 = 26 hours).

## 3 RAPID DESIGN OF SEQUENCES FOLDING INTO BOTH DE NOVO AND NATURAL SCAFFOLDS

For the goal of generating a protein sequence $x$ for a given target backbone $Y$, we would like to sample a sequence with high likelihood from:

$$x \sim p(x|y = Y) \tag{4}$$

which can be expressed with Bayes theorem as:

$$p(x|y = Y) = \frac{p(x)p(y = Y|x)}{p(y = Y)} \propto p(x)p(y = Y|x) \tag{5}$$

Where, given a fixed target, $p(y = Y)$ is constant. Here, $E_{LM}$ serves as a proxy of $p(x)$, representing the likelihood of the sequence. Instead, $E_T(x^\tau)$ approximates $p(y = Y|x)$, the probability of sequence $x$ folding into a structure $Y$. Resulting in the energy function of our state:

$$E(x) = \lambda_1 E_{LM}(x) + \lambda_2 E_T(x^\tau) \tag{6}$$

We use two datasets for fixed scaffold design. The first is a dataset of 37 artificially designed *de novo* proteins collected in Verkuil et al. (2022), spanning a range of lengths, folds, and *de novo* design methods. This set was used to demonstrate LM-design, a similar approach using MCMC on ESM2-derived energy functions where the $E_T(x^\tau)$ term was instead replaced by a term comparing a predicted contact map distogram to one extracted from the ground truth 3D template. The second is a dataset of 57 diverse CATH domains, subset from Ma & Bethel (2025) as those with protein length between 60 and 150, for which the authors used ProteinMPNN (Dauparas et al., 2022) to generate 160 designs per template. ProteinMPNN is a graph deep learning model specialised for the inverse folding task, trained explicitly on 3D structures to recover sequences for fixed backbones. The former dataset consists of proteins designed by humans without any evolutionary traces, and which contain structural peculiarities (unnatural binding pockets, shorter beta-turns etc.) not seen in natural proteins. The latter dataset, on the other hand, are evolutionarily honed functional folds often considered to have more complex folding and energy landscape than the typically stable designed folds. Unlike prior methods, we employ no 3D structural inputs; instead, amino acid sequences are converted into TEA sequences to serve as templates for generating 132 designs per template scaffold.

For the *de novo* scaffold dataset, TEA-LEAVES generates sequences that closely match target structures according to AlphaFold2 (Jumper et al., 2021) in single sequence, no template mode (using the model with the best pTM out of 5 models, hereafter referred to simply as AlphaFold2). Figure 2A illustrates the distribution of RMSDs and pTMs. Overall, 29/37 scaffolds have at least one successful design (success defined as $>0.7$ pTM and $<2.5$ Å RMSD), and 10% generated designs are successful, though this varies per scaffold. Crucially, TEA-LEAVES converges to these minima in just 30,000 ESM2 passes, providing a significant speed advantage over the 170,000 steps (340k ESM2 passes, $>10$hr/protein) required by LM-design (Verkuil et al., 2022) for similar results. In general, longer templates likely require more simulation steps as the number of mutations sampled per residue is lower, but this is left for future exploration. For the natural scaffold dataset, both ProteinMPNN and TEA-LEAVES performed poorly with AlphaFold2 (Supplementary Figure S1A), indicating that these scaffolds could be regions of fold space where AlphaFold2 cannot effectively be used as a scoring oracle. TEA-LEAVES produces comparable designs to ProteinMPNN for many scaffolds when using ESMFold as the scoring oracle (Supplementary Figure S1B), with 16 scaffolds having at least one successful design.

Successful TEA-LEAVES designs for the same template have high diversity (Figure 2B), demonstrating that each random Markov chain results in a unique solution to the design task. They also have low sequence identity to the template sequence (Figure 2C), demonstrating that the design

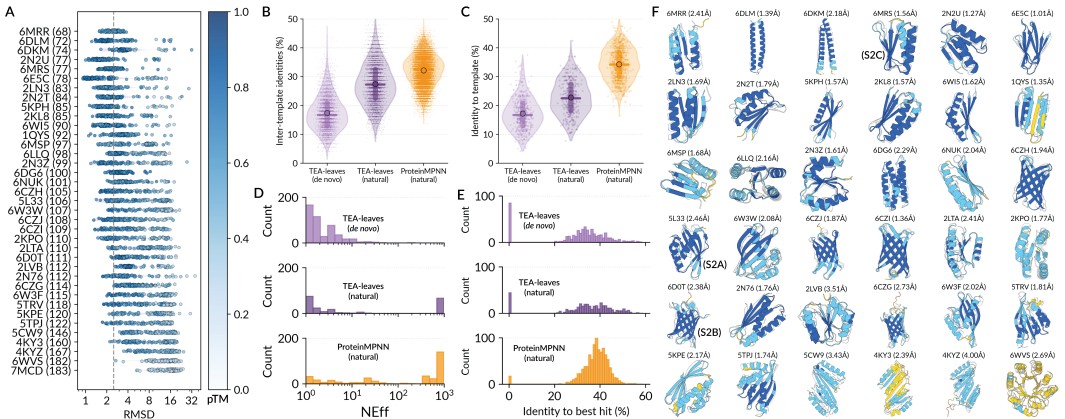

Figure 2: **Fixed backbone design A)** Distributions of RMSDs for designs based on *de novo* scaffolds predicted with AlphaFold2 in single sequence no template mode, coloured by pTM. **B-E)** For all successful designs (TM > 0.7 and pTM > 0.7, AlphaFold2 oracle for *de novo* and ESMFold oracle for natural scaffolds) for each of TEA-LEAVES on the *de novo* scaffolds, TEA-LEAVES on natural scaffolds, and ProteinMPNN on natural scaffolds, distributions of within-template design diversity (B), sequence identity to the template (C), number of effective sequences across UniRef90, BFD and MGnify (D), and Sequence identity from the local alignment to the best hit across UniRef90, BFD and MGnify (E). **F)** One designed structure per *de novo* scaffold, colored by AlphaFold2 pLDDT, with the template shown in white and backbone RMSD to the template labelled. Those with trajectories shown in Supplementary Figure S2 are indicated.

process produces novel solutions unrelated to the template sequence, and also that TEA-conversion effectively removes input amino acid sequence bias. Unlike ProteinMPNN designs, a significant fraction of successful TEA-LEAVES designs have a NEFF (number of effective sequences) (Haghani et al., 2025) of 1 when running the full JackHMMER-based AlphaFold2 MSA search pipeline (Figure 2D, 36% of successful designs have NEFF = 1 for TEA-LEAVES on *de novo* scaffolds, 21% for TEA-LEAVES on natural scaffolds and 3% for ProteinMPNN on natural scaffolds), indicating no significant hits across UniRef90 (Consortium, 2019), BFD (Steinegger & Söding, 2018), and MGnify (Richardson et al., 2023). Overall, TEA-LEAVES produces successful designs with very low to undetectable sequence identity compared to the natural protein space (Figure 2E). For the *de novo* scaffold set, the coverages for the detected hits are also low, with an average of 60±19%, indicating motif-level similarity rather than full protein. **Thus, with no structural input and in a matter of minutes, we generate multiple diverse and completely *de novo* designed proteins predicted to fold with high confidence by high-resolution structure prediction oracles into a target structure.**

Finally, Supplementary Figure S2 depicts MCMC trajectories for three *de novo* scaffold designs, underscoring the rapid convergence of our method to low RMSDs compared to previous approaches (Verkuil et al., 2022; Hie et al., 2022). The template energy $E_T(x^\tau)$ drops sharply within the first 5,000 steps, tracking closely with improved TM scores. However, achieving high confidence requires the simultaneous minimisation of the language model prior $E_{LM}(x)$, validating the synergy of our composite energy function.

## 4    UNCONDITIONAL GENERATION OF NOVEL SEQUENCES AND STRUCTURES

The problem of unconditional generation can be defined as sampling sequences $x$ and their associated structures $y$ from a high joint probability distribution:

$$x, y \sim p(x, y) = p(x)p(y|x) \tag{7}$$

In our framework, the energy term $E_{LM}(x)$ serves as a proxy for the sequence prior $p(x)$, while $E_H(x^\tau)$ represents the likelihood of a structure given a sequence. Initial experiments using these terms revealed a sampling bias toward long helical structures with low-entropy TEA sequences. To

counteract this and bias the sampling towards a more heterogeneous landscape, we introduced the diversity term $E_D(x^\tau)$, resulting in the energy term:

$$E(x) = \lambda_1 E_{LM}(x) + \lambda_2 E_H(x^\tau) + \lambda_3 E_D(x^\tau) \tag{8}$$

To evaluate the method's capacity for unconditional design, we generated 5,000 proteins of length 100 using the defined energy term and modelled the resulting sequences with AlphaFold2. Under these criteria, 42% of the designs achieved a pLDDT above 70, with the full distribution illustrated in Figure 3A. We subsequently restricted the dataset to these 2,129 high-confidence sequences for further characterisation. To assess the heterogeneity of the generated sequence and fold space, we performed clustering analyses using MMseqs2 and Foldseek with default "easy-cluster" parameters. Sequence clustering yielded no clusters, with all 2,129 sequences remaining as singletons, indicating high sequence diversity. Structural generation was, as expected, less heterogeneous, with the clustering resulting in 526 singletons and 237 clusters. These results confirm that our method effectively navigates a diverse sequence and structural landscape rather than collapsing into a limited set of motifs. The secondary structure composition, detailed in Figure 3B, further demonstrates the model's ability to generate varied combinations of structural elements. Finally, we assessed the novelty of our designs relative to the natural protein universe. Sensitive JackHMMER searches against UniRef90, MGnify and BFD resulted in the NEFF distribution shown in Figure 3C, demonstrating the sequence novelty of our designs, with 51% of designs having a NEFF of 1. Even in cases with hits, the sequence identity to the best hit is low, as shown in Figure 3D, highlighting their divergence relative to known protein families. A Foldseek query with a coverage threshold of 90% against the PDB revealed that 36% of our designs do not have a structural hit with TM-score greater than 0.5, demonstrating structural novelty.

**Overall, the unconditional generation process yields structurally plausible sequences (as measured by the AlphaFold2 oracle) characterised by significant novelty and structural diversity.** Though the pLDDT distribution is as yet worse than that obtained by LM-design for unconditional generation, the authors used $4 \times 170,000$ simulation steps (compared to the 30,000 used here) and a blocked Gibbs sampling approach to reach high pLDDT designs (Verkuil et al., 2022). Both options, increasing the number of simulation steps and alternating between free and templated design, along with better optimisation of energy weights and scheduler parameters, are available to TEA-LEAVES indicating that more performance can be gained for this task.

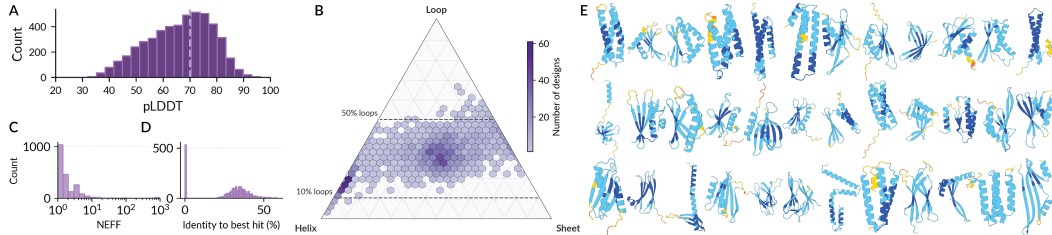

Figure 3: **Characterization of the unconditional design landscape.** **(A)** Distribution of pLDDT scores for 5,000 unconditional designs as predicted by AlphaFold2 in single sequence mode. The dashed line indicates the quality threshold of 70 used for selecting designs for the remaining panels. **(B)** Secondary structure composition (Helix, Sheet, and Loop) of the generated designs, showing a diverse topological distribution. **(C)** Distribution of the Number of Effective Sequences (NEFF) obtained from JackHMMER searches against UniRef90, MGnify, and BFD, highlighting sequence novelty. **(D)** Sequence identity to the closest natural homolog (0 if no hit) for each design **(E)** Representative gallery of 100-residue designs spanning a broad range of folds and topologies.

## 4.1 OUTLOOK

It is remarkable that complex 3D folds can be generated via simple 1D optimization of a 20-letter structure-inspired alphabet, suggesting that apparently local motifs can consistently guide global folding. While experimental validation is pending, our designs match the *in silico* metrics of experimentally verified proteins from LM-design (Lin et al., 2022) and achieve high confidence from AlphaFold2 and ESMFold, indicating a strong likelihood of solubility and folding.

While AlphaFold2 and to a lesser extent ESMFold have shown discriminatory power in distinguishing designability (Korbeld et al., 2025), both are known to have large areas of fold and sequence space where they are unable to predict known stable protein structures (Reidenbach et al., 2025). An approach such as TEA-LEAVES, making use of sequence-only models with very little structural guidance, may access a different slice of the fitness landscape from the current generation of *de novo* protein design methods. TEA-LEAVES and its adaptations could also serve as a powerful complementary tool to augment existing structure-based approaches, functioning as a generator of diverse, structurally viable starting points, constraints, or scores for more computationally intensive structural pipelines.

Although the current approach does not explicitly target protein function, the framework is highly extensible. Since ESM2 embeddings are computed at every step, lightweight prediction heads for properties like thermostability or binding, properties which pLMs are known to have predictive capacity for (Chu et al., 2024; Leclercq & Droit, 2025; Weissenow & Rost, 2025), can be integrated into the energy function with minimal overhead. This opens the door to multi-objective design, allowing for contact-based constraints or activity-guided enzyme engineering. Furthermore, the success of simple MCMC sampling to find diverse minima suggests that TEA could enable more advanced generative optimisation and design methods, including genetic algorithms, gradient-based optimisation, inverse folding, and discrete diffusion.

## 5 CONCLUSION

We have demonstrated that effective *de novo* protein design does not strictly require structure or contact map prediction oracles in the optimisation loop. By leveraging a discrete structural proxy derived from protein language models, we enabled random mutagenesis MCMC to rapidly navigate the protein landscape. Our results show that this approach successfully generates sequences that fold into both fixed targets and novel unconditional folds, passing rigorous *in silico* designability criteria while maintaining low to undetectable sequence identity to natural homologues. Ultimately, this framework holds promise for a computationally efficient and orthogonal alternative to current generative methods, potentially granting access to novel areas of design space.

### MEANINGFULNESS STATEMENT

Modern protein design techniques focus on geometric constraints rather than purely sequence-based logic. Our work shifts the focus toward meaningful latent representations, demonstrating that complex 3D folds can be generated by optimizing a structure-aware yet structure-independent alphabet derived from protein language models (pLMs). We show that pLMs can rapidly navigate the entire sequence landscape compatible with a given fold, capturing viable sequences that nature has yet to explore.

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

# A  APPENDIX

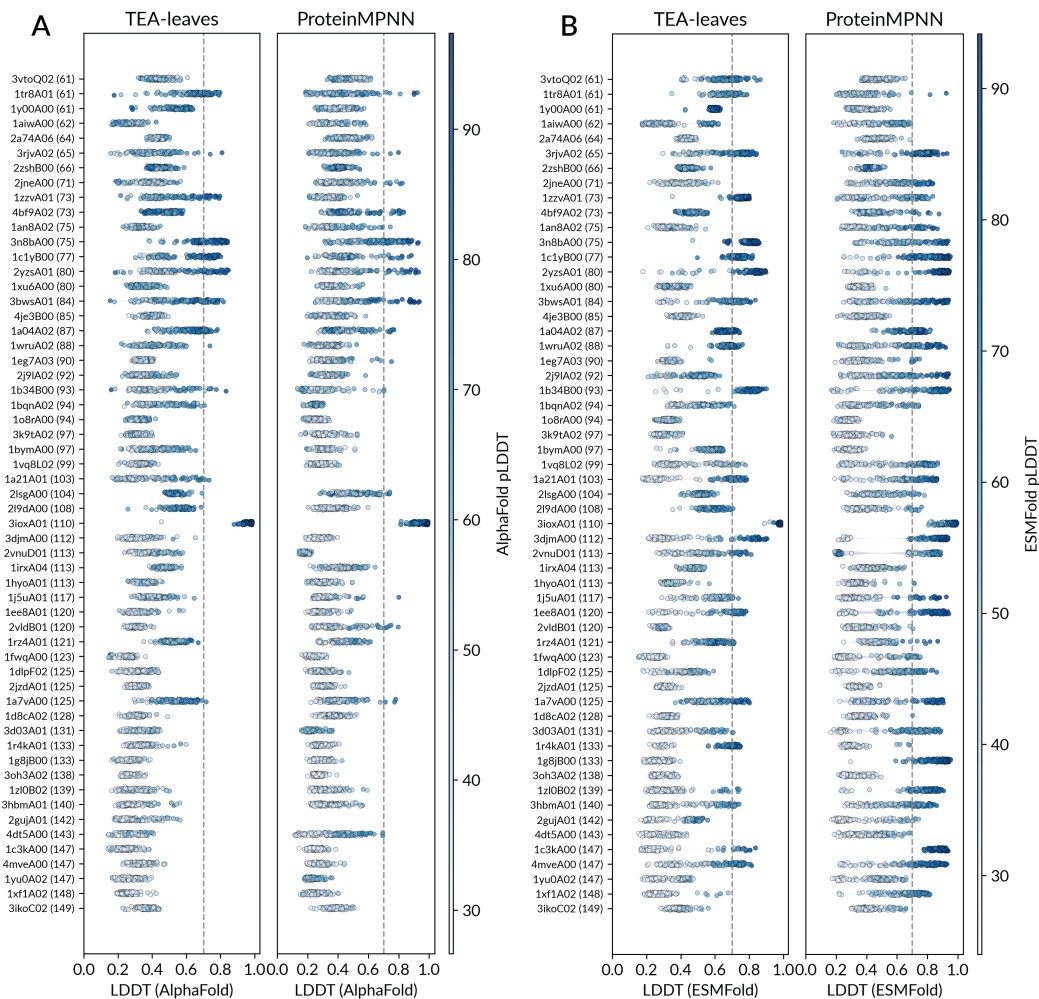

Figure S1: **Natural scaffold design.** Distributions of LDDTs vs. natural scaffold templates for 132 designs each from TEA-LEAVES and ProteinMPNN, coloured by pLDDT predicted with **A)** AlphaFold2 in single sequence no template mode, and **B)** ESMFold.

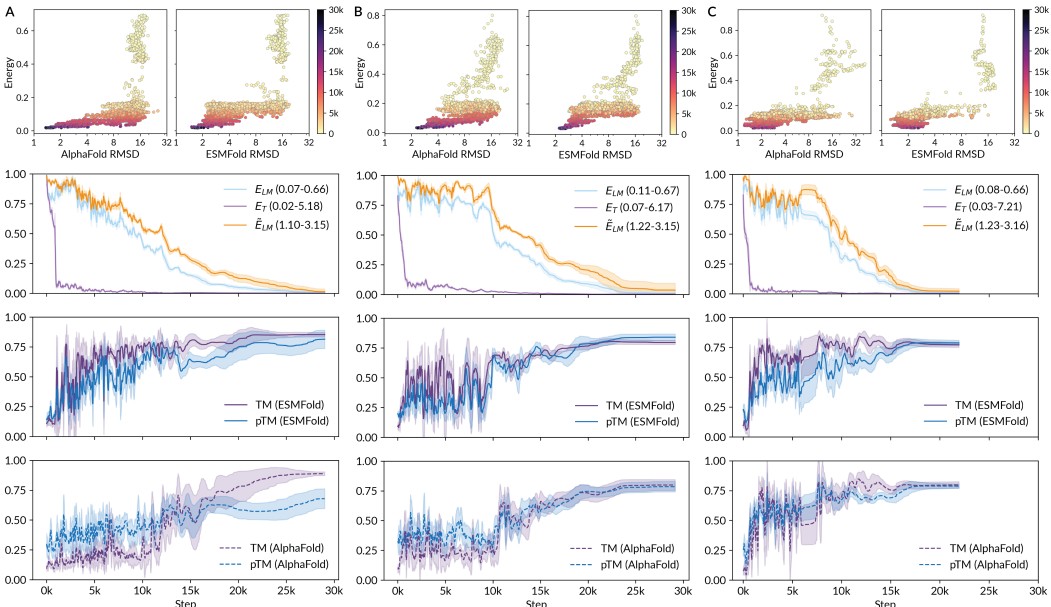

Figure S2: **Selected trajectories.** Depictions of various energy functions and oracle metrics across one trajectory each for **A)** 5L33 **B)** 6D0T and **C)** 6MRS (see Figure 2F for structure illustrations). Each panel contains energy vs. AlphaFold and ESMFold RMSDs to template coloured by step, gaussian-smoothed depiction of energies, ESMFold metrics and AlphaFold metrics across simulation time. Energy values are normalised to be between 0 and 1 to enable comparison, with minimum and maximum values displayed in the legend. $\tilde{E}_{LM}$ represents the masked marginals formulation of pseudo-log likelihood, which takes $N$ passes through ESM2.

