# OpenReview forum: "Reading TEA leaves for de novo protein design"
_ICLR.cc/2026/Workshop/LMRL — ICLR 2026 Workshop LMRL Poster_

### Official Review · Reviewer_32En · 2026-02-23
**Review for “Reading TEA leaves for de novo protein design”**

**Rating:** 9
**Confidence:** 4

**Review:**

Summary:

This paper introduces TEA-LEAVES, an energy-based MCMC framework for de novo protein design. By combining a pLM prior with TEA as a structural proxy, the method performs structure/template-guided and unconditional sequence design within an optimization loop without requiring expensive structure prediction. The authors demonstrate rapid convergence relative to prior MCMC approaches, validate structure conditioning in silico, and generate diverse sequences with low sequence identity to natural proteins.

=======================================

Strengths:

1. Overall framework is a simple, yet effective sampling method and achieves convergence substantially faster than LM-design.

2. TEA use as a structural likelihood proxy is a novel and efficient method for steering structure conditioning

3. The NEFF and Foldseek analyses support the claim that TEA-LEAVES generates structurally plausible yet sequence-divergent designs. In several settings, the approach compares favorably to ProteinMPNN in terms of novelty.

=======================================

Weaknesses:

1. While prior work on TEA demonstrates correlation between TEA entropy and structural confidence metrics, the manuscript would benefit from clarifying the extent to which TEA cross-entropy constitutes a calibrated structural likelihood versus a heuristic compatibility score.

2. It would strengthen the paper to include ablations examining the weighting of energy terms:

	a. Sensitivity to the λ weights across tasks

	b. The impact of the diversity term E_D, including the choice of k-mer size (k=3).

	c. The balance between E_LMand E_Tin de novo/natural scaffold design.

These components appear important, particularly for unconditional generation, but their relative contributions are not systematically evaluated.

3.  The work conceptually reminds me more of methods such as Chroma. How do benchmark metrics such as NEFF, Foldseek, or diversity coverage compare?

4. One additional method to show unconditional diversity (and related to above) is using SHAPES [1]

[1] Lu, T., Liu, M., Chen, Y., Kim, J., & Huang, P. S. (2025). Assessing generative model coverage of protein structures with SHAPES. Cell Systems, 16(8).

5. The manuscript reports 30k steps for length-100 proteins. It would be helpful to clarify how the required number of steps scales with sequence length and whether convergence properties degrade for longer proteins.

6. Given the role of Shannon entropy use by TEA for structural prediction proxy, could entropy-based filtering be used explicitly to identify high-confidence designs prior to structural oracle evaluation?

7. It may be useful to showcase the Foldseek analysis for sequence novelty as a SI figure.

8. Minor comment: Missing period “expensive Previous” line 039. The name “TEA-LEAVES” appears without formal introduction; consider adding a sentence such as: “We propose TEA-LEAVES, a framework…”

=======================================

Overall:

Exciting and computationally elegant idea that leverages discrete structural representations for efficient design. In silico results look very promising and would be interesting to see 1) experimental validation and 2) functional guidance. Additionally, would be interesting to see adaption to multimers and binder design.

---

### Official Review · Reviewer_ButE · 2026-02-25
**De Novo Protein Design via a Structure-Inspired Alphabet (TEA) and MCMC Sampling**

**Rating:** 7
**Confidence:** 4

**Review:**

### **Summary**

This paper introduces TEA-LEAVES, a *de novo* protein design framework based on Monte Carlo sampling guided by energy functions derived from a structure-inspired alphabet (TEA) computed from protein language model embeddings. The key idea is to avoid slow structure prediction or contact-map oracles during optimization (as seen in prior work) by operating in a transformed 20-letter alphabet called TEA (introduced in prior work), which captures latent structural information from ESM2 embeddings at minimal computational cost.

The method formulates protein design as an energy-based MCMC process using the Metropolis algorithm. At each step, a random mutation is proposed and accepted based on the change in a composite energy function. This energy combines a language model prior encouraging plausible sequences, a template-based structural likelihood term encouraging compatibility with a target backbone (for fixed scaffold design), a structural confidence term derived from TEA entropy, and, for unconditional generation, a diversity term that discourages repetitive structural motifs.

The authors evaluate fixed scaffold design on two datasets: 37 artificially designed *de novo* proteins and 57 natural CATH domains. For *de novo* scaffolds, 29 out of 37 templates yield at least one successful design according to AlphaFold2-based criteria, with convergence achieved in substantially fewer language model passes than prior MCMC-based approaches. In an unconditional setting, 5,000 length-100 sequences are generated, of which 42% achieve pLDDT greater than 70, with clustering analyses suggesting substantial sequence and structural diversity.

Overall, the work presents a computationally efficient and conceptually simple alternative to structure-based optimization loops. The use of a structure-informed alphabet to guide MCMC sampling is novel and interesting, though validation relies entirely on structure prediction oracles. Additionally, despite the clear efficiency improvements compared to prior approaches, this methodology does appear to be compute-intensive, raising concerns about widespread adoptability.

### **Strengths and Weaknesses**

### **Strengths**
- Clear motivation: replacing slow structural or contact-map oracles in MCMC-based protein design.
- Clever use of TEA, a structure-informed alphabet derived from pLM embeddings, to encode fold-level information at low computational cost.
- Well-defined energy formulation grounded in probabilistic intuition via Bayes’ theorem.
- Significant reduction in sampling steps compared to prior LM-based MCMC approaches.
- Demonstrated ability to design sequences for both artificial and natural scaffolds.
- High sequence diversity in both fixed and unconditional design settings, supported by clustering analyses and low identity to templates.
- Computational efficiency through quantized ESM2 and batched operations.

### **Weaknesses**
- The claim of “no structural input” is technically true in that no 3D coordinates are used during sampling, but structural information is implicitly encoded via TEA templates derived from known structures. Clarifying this would improve conceptual transparency.
- Success is defined entirely through structure prediction oracles (AlphaFold2/ESMFold), which are known to have biases and blind spots in fold space.
- The statement that designs are produced “in a matter of minutes” is somewhat misleading given that a single length-100 design takes ~24 minutes and large-scale experiments require hundreds of GPU hours.
- For natural scaffolds, performance depends strongly on which oracle is used, raising questions about robustness across evaluation methods.
- Claims that TEA conversion removes sequence bias would benefit from explicit controls.
- Unconditional generation achieves 42% pLDDT > 70, but relatively few designs reach very high confidence (>90), suggesting room for improvement.
- Experimental validation is absent; all claims rely on *in silico* metrics.
- Energy weights and annealing schedules are empirically chosen and fixed across experiments; sensitivity analyses are not presented.
- Certain terms could be explained more clearly, such as NEFF.
- The method is limited as protein length increases past 100, raising concerns about generalizability to diverse design tasks.
- The method's applicability to designing intrinsically disordered regions (if possible) is not addressed.

### **Final Assessment**

TEA-LEAVES presents an elegant and computationally efficient alternative to structure-driven protein design loops. By combining a language model prior with a discrete structural proxy derived from embeddings, the authors demonstrate that simple MCMC sampling can navigate protein sequence space far more rapidly than prior pLM-based approaches. The method is conceptually appealing and practically lightweight.

However, the evaluation depends entirely on structure prediction oracles and synthetic metrics of novelty. While the results are promising, experimental validation and broader benchmarking would be needed to fully substantiate the claim that this approach opens genuinely new regions of design space. Additionally, while this approach is far more efficient than prior methods discussed in the text, it still requires heavy GPU usage. Overall, this is a creative and technically interesting contribution that advances fast sequence-based protein design.

---

### Meta-Review · Area_Chair_prxw · 2026-02-27

**Recommendation:** Accept (Poster)
**Confidence:** 5

**Metareview:**

Accept.

---

### Decision · Program_Chairs · 2026-03-02

**Decision:**

Accept (Oral)

**Comment:**

Please see the meta-review.